# Efficient Test-time Scaling via Iterative Deepening

## Abstract

Recent reasoning models, such as OpenAI's O1 series, have demonstrated exceptional performance on complex reasoning tasks and revealed new test-time scaling laws. Inspired by this, many people have been studying how to train models to achieve effective self-evaluation and self-correction to further enable the scaling paradigm. However, less studied is how to efficiently scale test-time compute from a fixed model, and this remains a challenge. In this paper, we focus on whether LLMs can benefit from matching the pattern of correct responses. Specifically, we explore how systematically triggering a model's self-correction mechanisms can improve performance on challenging reasoning tasks. To this end, we propose a novel iterative deepening sampling algorithm framework designed to enhance self-correction and generate higher-quality samples. Through extensive experiments on Math500, AIME, and GPQA-diamond benchmarks, we demonstrate that our method achieves a higher success rate on difficult tasks and provide detailed ablation studies to analyze its effectiveness across diverse settings.

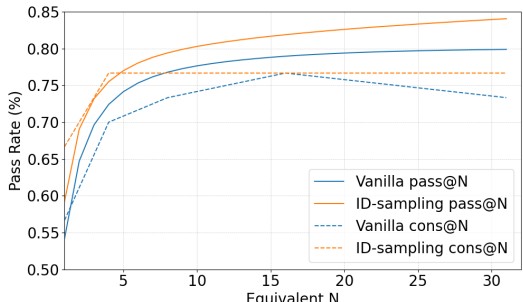

Figure 1: Comparison of our proposed ID-sampling to vanilla sampling on Qwen3-8B on AIME-25. The x-axis is the equivalent number of samples after considering the additional cost from ID-sampling.

## 1 Introduction

Since ChatGPT, large language models (LLMs) have been a rapidly evolving domain that tries to solve problems beyond traditional language tasks like summarization or question answering Chen et al. (2023); Yao et al. (2023); Chen et al. (2024b;d). Significantly, the newly released OpenAI O1 has introduced its new paradigm of test-time scaling, which shows strong capability in complex problem-solving through its detailed reasoning steps before outputting the final answer Jaech et al. (2024). Since then, many researchers have studied how to replicate success from an open-source perspective and how to train models that are even better at efficiently solving problems that still remain unsolvable by the current LLMs Huang et al. (2024); Zeng et al. (2024); DeepSeek-AI et al. (2025). One key finding is that through reinforcement learning itself, LLM can spontaneously learn to self-evaluate and self-correct from time to time. However, there is no clear conclusion on whether self-evaluation is triggered often enough.

Figure 2: An illustration of our method, Iterative Deepening Sampling (ID-sampling).

On the other hand, while training is the primary focus recently, it remains uncertain whether one could more efficiently scale test-time compute from a fixed model without additional changes in training or fine-tuning. Moreover, a more efficient sampling algorithm can not only enhance inference-time efficiency but also facilitate the generation of high-quality synthetic data, which can be leveraged to train the next generation of models and evaluate their performance Guan et al. (2025).

Recent work has investigated the differing patterns of correct versus incorrect responses. Notably, while o1-like models tend to achieve higher performance when provided with greater computational resources Huang et al. (2024); DeepSeek-AI et al. (2025), other studies have found that longer responses can actually reduce accuracy, a phenomenon often referred to as overthinking Marjanović et al. (2025). Given these distinct response patterns within the same model, a fundamental question arises: can we improve the test-time performance of LLMs by guiding generation to mimic the patterns of correct responses?

In this paper, we tackle the challenge of efficient test-time scaling using a fixed reasoning model without additional training, applied to complex reasoning tasks such as mathematical problem solving. Specifically, inspired by a novel observation on the positional distribution difference of linguistic markers that appear in correct and incorrect responses, we focus on manually inserting self-reflection triggers like "Wait" during the generation process to improve the pass rate of a fixed model. To achieve this, we propose Iterative Deepening Sampling (ID-Sampling), a novel algorithmic framework that iteratively increases the sampling budget following a geometric progression, while incorporating self-reflection mechanisms at each expansion step. We theoretically demonstrate that ID-Sampling effectively balances computational efficiency and response quality, ensuring that the budget is not excessively wasted while still improving model performance. We evaluate ID-Sampling on a few challenging reasoning benchmarks, and demonstrate its effectiveness in Best-of-N sampling and majority voting settings on various reasoning models. Additionally, we provide an ablation study analyzing how the rate of budget increase per iteration impacts both pass rate and inference time. Our results highlight the potential of ID-Sampling as a scalable approach for improving LLM reasoning performance through adaptive self-reflection mechanisms.

## 2 RELATED WORKS

There are two primary directions for test-time scaling, namely scaling on multiple responses, and scaling the reasoning steps in a single response.

To efficiently scale to multiple responses, researchers have proposed both aggregating independent samples and employing structured methods such as tree-search-based architectures. While there have been many works on tree-search strategies early on Zhang et al. (2023); Liu et al. (2023); Hao et al. (2023); Chen et al. (2024a); Zhou et al. (2023); Zhang et al. (2024a;b), researchers are putting more attention on aggregating independent responses given that it is hard to create a good process reward model to accurately estimate the quality of a partial solution DeepSeek-AI et al. (2025). Specifically in this direction, researchers mostly rely on two main classes of aggregating strategies, namely self-consistency Wang et al. (2022) and Best-of-N Snell et al. (2024); Wu et al. (2025). There are many works that focus on this setting to make it more efficient and effective. For example, Sun et al. Sun et al. (2024) proposed to use speculative rejection in BoN to reject bad responses through early scores. Chen et al. Chen et al. (2024e) proposed to use extremely high temperature on the first token to greatly improve the Best-of-N performance on math and coding tasks.

To scale within a single response, researchers have first introduced an intermediate step by the so-called chain-of-thoughts Wang & Zhou (2024). Then, starting with Self-Refine Madaan et al. (2023), there are many works that studied how to effectively use an LLM to give themselves their

own evaluation and reflection Yao et al. (2023); Shinn et al. (2023), and study how this reflection-based mechanism can be adapted to different applications Chen et al. (2023); Gou et al. (2023); Chen et al. (2024c). While there have previously been a lot of discussions on whether LLM can actually self-evaluate and self-correct themselves Huang et al. (2023); Chen et al. (2024f); Verma et al. (2024); Wang et al. (2025b), recent studies have shown that by training with high-quality data that involve such a process, LLM can actually achieve a useable level of capability and help self-correct in the process Huang et al. (2024); Zeng et al. (2024). Inspired by the recent success of Deepseek-R1 DeepSeek-AI et al. (2025), people have realized that scaling within a single-response can be efficiently achieved through reinforcement learning, and many follow-up works have also studied how to learn such scaling capability more efficiently Wang et al. (2025a). In this paper, we manually inject *trigger sentence* in the middle of generation, which is similar to the recent work s1 Muennighoff et al. (2025). However, our work focuses on injecting at an early stage in the thinking process, while they focus on injecting when the current response is completely finished for budget forcing. As we later show in our experiments, this design difference leads to a substantial divergence in results: their approach requires further training, whereas ours provides benefits directly with existing models.

While the research in the two search strategies is mostly separate, their methods are orthogonal and can be used together to make the sampling process more efficient Snell et al. (2024), and some recent research has also studied how to balance the two, and has shown that scaling on the number of responses can be more efficient than scaling within a single response Wang et al. (2025a); Marjanović et al. (2025); Sadhukhan et al. (2025). In this paper, we focus on the intersection of both scaling strategies and study how to strategically trigger the self-correction capability of LLMs more efficiently. We take into account the additional cost of scaling within one response by measuring the total wall-clock time used, and calculate the equivalent $N$ when comparing the answers as needed.

## 3 PRELIMINARIES

### 3.1 BEST-OF-N SAMPLING

For reasoning-intensive tasks such as mathematical problem-solving and coding, Best-of-N sampling is one of the most widely used strategies for data generation. This approach involves sampling N outputs from the same model using predefined sampling parameters — typically with a higher temperature than single-sample settings — followed by a selection process to determine the best response. The selection criteria depend on the intended use of the samples. During training, responses are typically evaluated using rule-based checkers for mathematical problems or online judges for coding tasks to identify correct answers within the sampled set. At test time, a reward model is often employed to score the generated responses, with the highest-scoring output selected as the final answer. This methodology effectively balances exploration and optimization, making it a fundamental component in enhancing the performance of LLMs on reasoning tasks.

The BoN sampling is a simple yet effective method that can be fully parallelized to enhance performance. Increasing N guarantees improved results during training when a ground-truth checker is available and generally leads to better performance at inference time, provided that the reward model is sufficiently accurate. However, if paired with a reward model that is not good enough, BoN might fail to scale efficiently and might even decrease its performance when more samples are included for aggregation. Therefore, getting a good reward model is necessary for good performance for BoN.

### 3.2 MAJORITY VOTING

Similar to BoN sampling, majority voting, also known as self-consistency (cons@n), provides an alternative approach for aggregating N different responses Wang et al. (2022). As the name suggests, this method involves generating N responses, counting the frequency of each unique answer, and selecting the most frequently occurring response as the final output. This approach leverages the inherent redundancy in multiple generations to improve robustness and reliability, making it particularly useful for tasks requiring high confidence in correctness.

Majority voting offers the advantage of aggregating responses without relying on a reward model. Additionally, although empirically less effective, it can be extended to a weighted version, where weights are assigned based on PRM scores or confidence estimations of the generated answers Wang

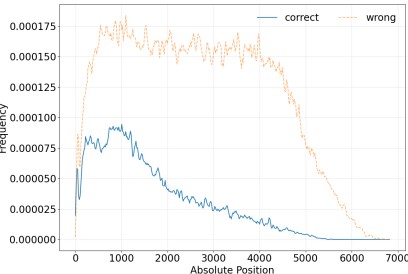

Figure 3: Positional frequency of thinking-related linguistic markers in Deepseek-R1 responses on the AIME-2024 dataset. A higher frequency indicates the likelihood of a given position containing such markers within a response. The plot is smoothed with a moving average for better visualization.

et al. (2022). While majority voting benefits from not requiring a highly accurate reward model, it faces challenges in identifying equivalent answers. For instance, it can be very hard to identify two code samples to be the same, and even in mathematical problems, expressions such as $\frac{1}{\sqrt{3}}$ and $\frac{\sqrt{3}}{3}$ are equivalent but must be recognized as such to ensure correct vote counting. A common solution in mathematical domains involves using symbolic-based checkers to compare answer pairs and identify equivalences. However, this process can be computationally expensive, requiring up to $O(N^2)$ comparisons.

## 4 ITERATIVE DEEPENING SAMPLING

### 4.1 OBSERVATION: FREQUENCIES OF LINGUISTIC MARKERS IN REASONING MODELS

Recent studies have highlighted that overthinking can negatively impact the performance of reasoning models Wang et al. (2025a); Marjanović et al. (2025). A key observation is that correct answers tend to be shorter and contain fewer linguistic markers associated with "thinking" behaviors.

Here, we present a novel perspective by analyzing not only the frequency but also the positional distribution of thinking-related markers within model responses. A complete list of these markers is provided in the appendix. As illustrated in Fig. 3, correct answers tend to exhibit fewer think markers, with a clear concentration in the early stages of the response. Their frequency drops sharply after the first 1000 tokens. In contrast, incorrect answers often contain self-corrections distributed throughout the response, with a noticeable decline only near the end. While this distinct distribution pattern is less informative than response length for classifying correctness due to its complexity, it is nonetheless a characteristic that we can mimic during generation.

### 4.2 METHODS

In this paper, we focus on efficiently scaling test-time computation by replacing the standard sampling algorithm with a more effective, custom-designed alternative. Given a fixed overall computational budget, an important challenge is determining how much additional budget should be allocated to refining a given prefix $x_0$ that the model has already sampled. Efficient budget allocation is crucial, as any saved resources can be redirected to increasing the number of N in Best-of-N sampling or deepening tree search, ultimately improving overall performance. Understanding this trade-off is key to optimizing both search efficiency and model output quality.

More specifically, the self-evaluation and self-correction process of the LLM can be manually triggered by introducing a predefined *trigger sentence*, such as "Wait! Maybe I made some mistakes! I need to rethink from scratch." or simply "Wait". In most cases, LLMs respond to this trigger by restarting their reasoning process and making self-corrections. Repeatedly inserting this sentence increases the overall length of the reasoning trajectory, potentially improving problem-solving accuracy by facilitating iterative refinement. In this paper, we propose a method for strategically placing a fixed *trigger sentence* at increasing intervals within longer contextual windows, aiming to balance computational efficiency and the effectiveness of self-correction.

---

**Algorithm 1** Iterative Deepening Sampling

---

1: **function** GENERATE_WITH_ID-SAMPLING(LLM, Dataset, $B_0$, $\gamma$, $B$):
2:   $Prefixes \leftarrow Questions\ from\ Dataset$
3:   $Budget \leftarrow B_0$
4:   **while** $Prefixes$ is not empty **and** $Budget \leq B$ **do**
5:     $NewPrefixes \leftarrow \{\}$
6:     $Outputs \leftarrow$ LLM.generate($input = Prefixes, max\_tokens = Budget$)
7:     **for** each $Output$ in $Outputs$ **do**
8:       **if** $Output$ is finished **then**
9:         LOGANSWER($Output$)
10:       **else if** $Output$ finished thinking **then**
11:         $Output \leftarrow$ LLM.generate($input = Output$)
12:         LOGANSWER($Output$)
13:       **else**
14:         $NewPrefix \leftarrow$ PADTRIGGERSENTENCE($Output$)
15:         $NewPrefixes$.append($NewPrefix$)
16:       **end if**
17:     **end for**
18:     $Budget \leftarrow Budget \times \gamma$
19:     $Prefixes \leftarrow NewPrefixes$
20: **end while**
21: **end function**

---

Motivated by the analysis in Section 4.1, we propose to introduce more *trigger sentences* at earlier stages and gradually reduce their occurrence as the response length increases. Suppose we have already used a budget of $b$ to generate a prefix $x_0$ and are now considering whether to immediately introduce a *trigger sentence*. Iterative Deepening (ID) sampling allocates an additional budget of $\gamma \times b$ before inserting the *trigger sentence* the next time, where $\gamma > 1$ is a tunable hyperparameter. This iterative process continues until reaching a maximum budget $B$, beyond which no further *trigger sentences* are introduced. If the reasoning process reaches a natural stopping point—i.e., a complete answer is generated within the allocated budget—the process terminates early. This is because generating a full response from scratch generally leads to more reliable outputs than attempting to refine an already complete solution. The complete procedure is outlined in Algorithm 1, where $B_0$ represents the initial budget. The function LLM.generate conducts the generation within a given budget and is adaptable to different tree-search strategies. The function PadTriggerSentence handles the insertion of *trigger sentences* while ensuring redundancy is minimized if necessary.

The definition and allocation of computational budget depend on the specific test-time scaling algorithm employed, leading to variations in implementation strategies. In this paper, we focus on the setting where N responses are sampled independently, and we have provided an illustration of ID-sampling in Fig. 2. Since response generation is independent and typically performed in parallel, the computational cost primarily depends on the total number of generated responses $N$ and their respective lengths. To manage computational efficiency, we define the budget as the maximum number of tokens generated in a given round, which corresponds to the $max\_token$ parameter in LLM serving engines such as vLLM Kwon et al. (2023). Additionally, to avoid inserting the *trigger sentence* mid-sentence, we extend generation until the completion of a reasoning step. Here, a step is identified by token splits such as '\n' or '\n\n'. This ensures that *trigger sentence* placement aligns with the basic logical structure of the generated response, preserving coherence and stability in the reasoning process. On the other hand, this also helps introduce some randomness in inserting *trigger sentence*, avoiding inserting them on exactly the same token at each run.

In implementations, the maximum generation token limit is a common parameter provided by modern serving engines such as vLLM or OpenAI's API services. Consequently, ID-sampling can still efficiently leverage the speedups offered by these engines. The pseudocode is provided in Alg. 1.

### 4.3 THEORETICAL ANALYSIS

A key challenge in ID-sampling is that budget control occurs before each manually triggered self-evaluation and self-correction step without explicitly analyzing the actual generated responses. This can lead to unnecessary iterations, potentially increasing computational costs. However, due to the design of our ID-sampling method, we establish important theoretical guarantees. In particular, we provide a bound on the total number of tokens generated before reaching the final answer, as formalized in the following theorem.

**Theorem 4.1.** *Suppose the final answer obtained through ID-Sampling needs a budget of $L$ in normal sampling without manual injection in the middle. Then the total number of additional budget used is no more than $\frac{\gamma * L}{\gamma - 1}$.*

**Proof Sketch** Note that the budget for each generation iteration follows a geometric sequence with common ratio $\gamma$. The theorem follows directly from a summation of this geometric series. A complete proof is provided in the appendix.

The theorem does not guarantee the quality of the generated responses or the alignment of the output distribution. Rather, it establishes that our method incurs no substantial additional computational overhead when generating a single response. Our intuition, however, is that by introducing *trigger sentences* at an early stage, ID-sampling can yield empirical gains in pass rate by biasing the model's output distribution toward higher-quality responses.

It is important to note that Iterative Deepening (ID) sampling does not impose any assumptions or constraints on the model's inherent self-correction or self-evaluation capabilities. In some cases, the model may naturally generate a response that already includes a *trigger sentence* before an explicit manual insertion. We observe that the built-in reasoning capabilities of recent state-of-the-art models, such as OpenAI-o1 Jaech et al. (2024) and DeepSeek-R1 DeepSeek-AI et al. (2025), significantly impact the effectiveness of ID-sampling. To better understand these effects, we conduct a comprehensive study using DeepSeek-R1, which we present in the experimental section.

## 5 EXPERIMENTS

### 5.1 EXPERIMENT SETUP

**Models and Datasets** In our experiments, we primarily target improvements in reasoning models where test-time scaling actually holds, while also reporting results for non-reasoning models as a point of reference. We use Llama-3.1-1B-Instruct Dubey et al. (2024) and Phi-4 Abdin et al. (2024) as examples for non-reasoning models, and DeepSeek-R1-Distill-Qwen-7BDeepSeek-AI et al. (2025), Qwen3-8B and Qwen3-32B Yang et al. (2025) for reasoning models. For reward models for BoN, we employ Qwen-2.5-Math-PRM-7B Zhang et al. (2025). We evaluate ID-sampling on three benchmark datasets: MATH-500 Lightman et al. (2023), AIME, and GPQA-diamond Rein et al. (2024). The first two are math-focused datasets, while GPQA-diamond is a science QA benchmark. Due to space constraints, results on GPQA-diamond are presented in the appendix.

**Baselines** We compare our approach against two settings: (i) vanilla sampling without any manual intervention, and (ii) S1 Muennighoff et al. (2025) without the training mentioned, which is simply appends word "wait" after the original reasoning process is completed twice.

**Evaluation** We report three key pass rate metrics across the datasets:

1. Best-of-N (BoN) – The accuracy when a reward model selects the best response from N generated samples.

2. Pass@N - The pass rate, which measures whether or not at least one of the N total responses is correct. We have used the unbiased estimation version Chen et al. (2021) with 40 total samples to address the statistical significance problem.

3. Majority Voting (cons@N) – The accuracy when responses are aggregated via unweighted majority voting, selecting the most frequent answer.

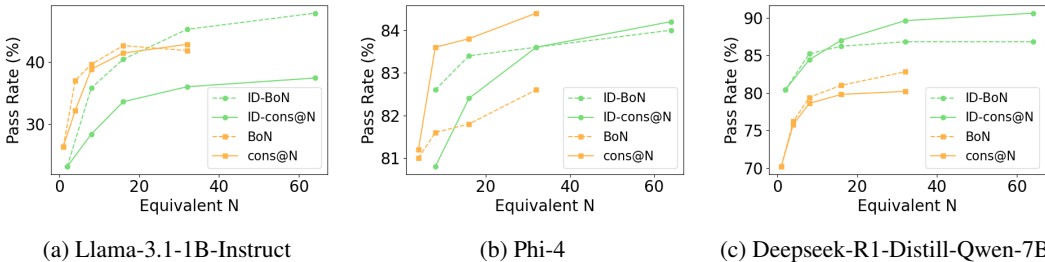

Figure 4: Math-500 dataset: Pass rate results for different models. The x-axis is the equivalent N after considering the extra time used by ID-sampling.

For both BoN and majority voting, a single aggregated answer is compared against the ground-truth solution to measure accuracy. We report BoN on Math500 datasets and for non-reasoning models on AIME datasets, given that its response length is within the context length limit of the reward models, and report Pass@n for reasoning models on AIME datasets because the normal response length is $> 12K$ and beyond the context length of popular reward models. [1]

For both math benchmarks, we employ symbolic checkers to ensure that all mathematically equivalent answers are accepted as correct.

**Parameters** For *trigger sentence*, we choose to use a whole sentence "Wait! Maybe I made some mistakes! I need to rethink from scratch." for non-reasoning models and a single word "wait" for reasoning models. For the word "wait", the natural generation behavior without ID sampling follows the same pattern shown in Fig. 3. We will provide a discussion about this choice later. By default, we use $\gamma = 2.0$ for ID-sampling, and we will later provide ablation study results on this. For ID-sampling with BoN, we set the initial budget $B_0$ as 256 tokens. For reasoning models, we set the thinking budget as 4096 for any model on Math-500 datasets, and 16384 for AIME datasets. If the model fully used the thinking budget, an end-of-think token (`</think>`) is padded, and an additional 1000 tokens are provided for the final answer to be provided. For non-thinking models, we set a fixed max sequence length of 4096. Due to the page limit, we leave other hyperparameters in the appendix.

## 5.2 RESULTS

### 5.2.1 MATH-500

Before evaluating the pass rate, we first analyze the runtime overhead of ID-sampling on Math-500 datasets. We observe that ID-sampling incurs approximately 1.6–1.9× the total wall-clock time compared to the baseline of vanilla sampling for non-reasoning models, and 1.1–1.3× for reasoning models. Given that this additional computational cost could instead be allocated to generating more responses and selecting the best one, we present our results in terms of an equivalent N. Specifically, if the original results correspond to Bo8, we report them as equivalent $N = 16$, as ID-sampling consistently completes within twice the runtime of the original method.

We present our results for different models on Math-500 in Fig. 4. We observe a notable difference in the effectiveness of ID-sampling for non-reasoning models based on the aggregation method. While ID-sampling yields performance gains with increasing sample size (N) under a Best-of-N (BoN) selection strategy, it consistently performs poorly when results are combined via majority voting. This suggests that for non-reasoning models lacking self-reflection capabilities, the patterns of correct versus incorrect responses differ in ways that ID-sampling fails to exploit.

By contrast, our primary focus is on reasoning models with test-time scaling capabilities. With the DeepSeek-R1-Distill-Qwen-7B, a strong reasoning model with built-in self-evaluation and self-correction, ID-sampling consistently outperforms vanilla sampling. This is because while stronger

---

[1]We have tried to use techniques like Yarn to extend the context length limit to a longer context. However, the model will lead to a consistent decrease rather than an increase when we increase the number of responses, and thus, we decided to just report Pass@n on longer context answers.

| $N$ | Vanilla | | ID-sampling | | $N$ | Vanilla | | ID-sampling | |
|---|---|---|---|---|---|---|---|---|---|
| | BoN | cons@N | BoN | cons@N | | BoN | cons@N | BoN | cons@N |
| 1 | 0.00 | 0.00 | **3.45** | **3.45** | 1 | 17.24 | 17.24 | **20.69** | **20.69** |
| 4 | 0.00 | 0.00 | 0.00 | **3.45** | 4 | 13.79 | 17.24 | **24.14** | **24.14** |
| 8 | 0.00 | 0.00 | 0.00 | **3.45** | 8 | 13.79 | 20.69 | **27.59** | 24.14 |
| 16 | **3.45** | 0.00 | 0.00 | **3.45** | 16 | 13.79 | **20.69** | **27.59** | 20.69 |
| 32 | 3.45 | 0.00 | **10.34** | **6.90** | 32 | 13.79 | **20.69** | **27.59** | 20.69 |

(a) Llama-3.1-1B-Instruct
(b) Phi-4

Table 1: AIME-24 dataset: Pass rate results for different models with different number of samples. The best results in each setting are highlighted in bold. Since performance saturates quickly on these weaker LLM models, meaning that even the highest pass rate with $N = 32$ does not exceed the $N = 1$ pass rate for ID-sampling, we omit the use of equivalent N in this setting.

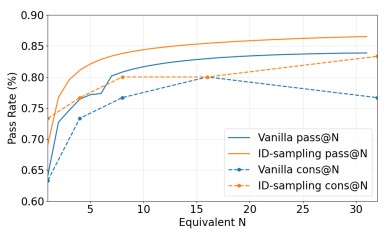

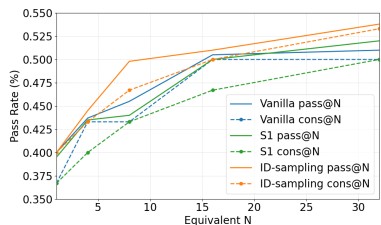

Figure 5: Qwen3-8B results on AIME-24.

Figure 6: Deepseek-R1-distill-Qwen-7B results on AIME-25.

models excel at self-correction, they remain suboptimal at determining when to initiate the self-correction process. Compared to earlier models, each *trigger sentence* has a more pronounced effect, allowing ID-sampling to correct errors that might otherwise persist without explicit intervention.

### 5.2.2 AIME

The AIME datasets (AIME-24 and AIME-25) are highly challenging benchmarks that have become widely adopted for evaluating reasoning models DeepSeek-AI et al. (2025); Huang et al. (2024). In this setting, reward models exhibit substantially lower accuracy. Accordingly, rather than reporting BoN, we adopt Pass@N and Cons@N as our evaluation metrics.

We first present the results for non-reasoning models in Table 1. We observe that ID-sampling can help improve the pass rate, and because of the fast saturation of pass rate, ID-sampling can even help the model to surpass its previous ceiling, i.e., $N = 1$ with ID-sampling is better than $N = 32$ with vanilla sampling. More importantly, for reasoning models, the wall-clock-time difference between vanilla sampling and ID-sampling becomes negligible. Specifically, for Qwen3 models, the time ratio remains within $1 \pm 0.02$, and for R1-distill models, it is within $1 \pm 0.1$ with $\gamma = 2.0$. As a result, the use of equivalent $N$ has little practical effect in this context. We present results on reasoning models in Fig. 5, 1, 6, 7, and Table 2. Across multiple models on the two most recent AIME

| n | 1 | 4 | 8 | 16 |
|---|---|---|---|---|
| Vanilla Pass@n | 67.8 | 81.6 | 85.9 | 88.4 |
| S1 Pass @n | 67.6 | 82.0 | **86.1** | 88.4 |
| ID-sampling Pass@n | **69.5** | **82.8** | **86.1** | **88.9** |
| Vanilla Cons@n | 56.7 | 76.7 | **80.0** | 83.3 |
| S1 Cons@n | 56.7 | 76.7 | 76.7 | 83.3 |
| ID-sampling Cons@n | **63.3** | **80.0** | **80.0** | 83.3 |

Table 2: Comparison between Vanilla sampling, S1, and ID-sampling on Qwen3-32B on AIME-25.

| | Rel. Time |
|---|---|
| Vanilla | 1.00 |
| ID(2.5) | 1.07 |
| ID(2.0) | 1.09 |
| ID(1.5) | 1.39 |

Table 3: Relative wall-clock time cost of ID-sampling for varying $\gamma$ values using DeepSeek-R1-Distill-Qwen-7B on the AIME-24 dataset.

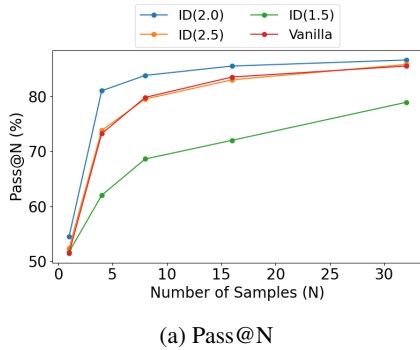 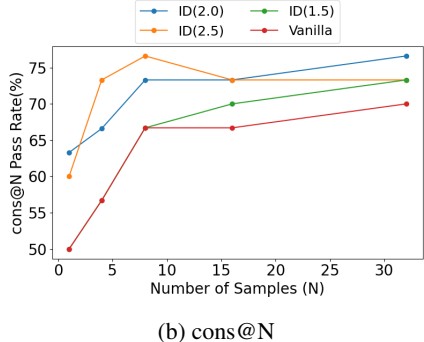

(a) Pass@N              (b) cons@N

Figure 7: AIME-24 Dataset: Pass rate results for DeepSeek-R1-Distill-Qwen-7B with different $\gamma$ for ID-sampling, compared to vanilla sampling. The numbers in brackets are the $\gamma$ used in ID-sampling. As different values of $\gamma$ result in significantly varying runtimes, as shown in Table 3, we again omit the use of equivalent $N$ in this analysis.

datasets, ID-sampling consistently outperforms both vanilla sampling and S1. This demonstrates the general applicability of our method, without requiring any assumptions about the specifics of post-training. By contrast, S1 only barely matches the performance of vanilla sampling on Deepseek-R1-distill-Qwen-7B, reinforcing the need for training as originally emphasized in the S1 paper. This is because S1 does not substantially alter the overall response pattern, particularly when measured by the positional frequency of linguistic markers, and therefore derives no benefit from it. For Qwen3-8B, which typically exhausts the full thinking budget, S1 has no opportunity to be applied, and its performance largely overlaps with vanilla sampling on both AIME-24 and AIME-25. For clarity, we therefore omit the S1 curve for Qwen3-8B models.

**Ablation Study** Given the stable performance gains of ID-sampling on AIME, here we conducted an ablation study to analyze the impact of the scaling factor $\gamma$ on ID-sampling. We present the pass rate results for DeepSeek-R1-Distill-Qwen-7B on AIME-24 in Figures 7a and 7b, and report the relative inference time for each setting in Table 3.

We find that adjusting $\gamma$ significantly impacts both performance and computational cost. In terms of runtime, $\gamma = 1.5$ yields the highest cost, whereas the other two settings remain within $1.1\times$ the wall-clock time of vanilla sampling. Regarding performance, $\gamma = 2.0$ consistently achieves the best results in terms of Pass@$N$, while $\gamma = 2.5$ occasionally outperforms in terms of Cons@$N$, though with a small margin. Overall, ID-sampling proves to be a more effective sampling strategy than vanilla sampling, provided that $\gamma$ is not too small, which would undermine our goal of emphasizing early-stage *trigger sentence* injection.

We observed a non-convex relationship between $\gamma$ and performance, even without accounting for time, with one of the best settings at $\gamma = 2.0$. We attribute this to the fact that ID-sampling aims to match the patterns of correct responses, but can mismatch when the *trigger sentence* is inserted too frequently, i.e. $\gamma$ is small. Overall, as most reasoning models right now share a similar reasoning pattern, we recommend $\gamma = 2.0$ as it strikes a favorable balance between performance and efficiency, and we have demonstrated its effectiveness through the extensive experiments presented above.

## 6 DISCUSSIONS

**Choice of Trigger Sentence** In our experiments, we use different *trigger sentences* for non-reasoning models and reasoning models. The difference is caused by the nature of the models. For non-reasoning models, a single word "wait" cannot trigger the self-correction process and can only introduce noise to the generation. For reasoning models, we found that for an unknown reason, adding the whole sentence will trigger the generation of an end-of-think token for a certain probability, and lead to an early stop in the generation sequence. While this does not always harm the performance, this is not something we want, as we want to use the *trigger sentence* to trigger

self-correction. Overall, as test-time scaling is highly correlated to reasoning models, a single word, "wait," will be sufficient as shown in our experiments.

**Limitations** Our proposed method also has several clear limitations. The most significant is that the performance of our method could significantly depend on the underlying model. While we have used our experiments to show that ID-sampling works on popular models, it is possible for some models to be incompatible with the algorithm, especially if the models are trained with losses that incorporate more than just the accuracy of the final answer. Moreover, incorporating *trigger sentences* into the generation process requires ID-sampling to invoke multiple generation steps. While this theoretically incurs no additional cost on the KV-cache, in practice, it can lead to increased inference time. To address this concern, we use equivalent $N$ in our experiments, which accounts for total wall-clock time, to enable a fair comparison across different sampling methods. Additionally, due to the need for multiple sampling steps, ID-sampling is currently tested only on open-source models. While the same idea can also be applied to black-box models through multi-round generations, this will introduce extra assistant tokens during generation, which may cause a slight distribution shift when applied to black-box models.

## 7 Conclusions

In this paper, we introduced Iterative Deepening (ID) Sampling, a simple yet effective algorithm for scaling test-time compute more efficiently than standard sampling. We showed that ID-sampling improves test-time performance on challenging reasoning tasks like math across diverse reasoning models. Our findings suggest that while current models possess strong self-correction capabilities, they remain limited in autonomously deciding when to invoke such mechanisms. Furthermore, our results highlight that guiding generation to mimic the patterns of correct responses can provide a benefit at test-time.

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

## A    DISCLAIMER OF LLM USAGE

Apart from being the subject of study, large language models were used solely to polish the writing in this paper. All substantive contributions are attributable to the authors.

## B    COMPLETE PROOF FOR THEOREM 4.1

Let $L_{total}$ represent the total generated length resulting from an iterative process. Let $l$ denote the length generated in the final iteration. We naturally have $l \leq L$, otherwise the generation will be finished earlier.

We aim to establish an upper bound for $L_{total}$. We analyze the contributions to the total length by considering the sequence of generated lengths in reverse chronological order. Assume that this sequence of lengths can be bounded above by a geometric sequence whose first term is $l$ and whose common ratio is $r = 1/\gamma$, where $\gamma$ is a constant greater than 1 ($\gamma > 1$).

Under this assumption, the total length $L_{total}$ is bounded by the sum of this infinite geometric series:

$$L_{total} \leq \sum_{k=0}^{\infty} l \left( \frac{1}{\gamma} \right)^k$$

Since it is assumed that $\gamma > 1$, the common ratio $r = 1/\gamma$ satisfies $0 < r < 1$. Therefore, the geometric series converges, and its sum is:

$$\sum_{k=0}^{\infty} l \left(\frac{1}{\gamma}\right)^k = \frac{l}{1 - \frac{1}{\gamma}}$$

We can simplify the denominator of this expression:

$$\frac{l}{1 - \frac{1}{\gamma}} = \frac{l}{\frac{\gamma-1}{\gamma}} = \frac{\gamma l}{\gamma - 1}$$

Thus, we have the bound:

$$L_{total} \leq \frac{\gamma l}{\gamma - 1}$$

Finally, using the given condition that $l \leq L$, we substitute $L$ to obtain the final upper bound for the total length:

$$L_{total} \leq \frac{\gamma l}{\gamma - 1} \leq \frac{\gamma L}{\gamma - 1}$$

Therefore, the total generated length $L_{total}$ is bounded above by $\frac{\gamma L}{\gamma-1}$.

## C    HYPERPARAMETERS

All experiments are conducted on an NVIDIA A100 GPU (80GB memory) using the vLLM serving engine Kwon et al. (2023). By default, we use a carefully selected set of hyperparameters and configurations to balance efficiency and performance:

- Based on the typical sampling budget used in training (10–40), we set the maximum number of N to 32 for sampling.

- All numbers are based on our implementations and the numbers are slightly different to the ones reported on technical reports due to the prompt used and the early cut-off based on the budget.

- By default, we use $temperature = 0.6, top\_p = 0.95$. For Qwen3, we additionally use $top\_k = 20$ as recommended by the official technical report.

## D    ADDITIONAL DISCUSSIONS

**Potential Extension to Search-based Test-time Scaling**    In this paper, our study primarily focuses on ID-sampling without integrating it into tree-search algorithms. However, we emphasize that our proposed framework can be directly extended to methods such as beam search and Monte Carlo Tree Search (MCTS). In these cases, the budget in Algorithm 1 can be directly interpreted as the computational budget in MCTS or the number of iterations in beam search. One challenge in this extension is that self-correction mechanisms reduce the reliance on early model outputs being correct, distinguishing the need for a new generation of PRMs from previously released ones. However, future researchers who develop or have access to more accurate PRMs that explicitly model the self-correction process can leverage our ID-sampling framework to enhance the reasoning capabilities of fixed models.

**Choice of Datasets**    In our experiments, we focus primarily on mathematical reasoning datasets, with results on a science QA dataset provided in the appendix. While test-time scaling techniques could, in principle, be extended to more general reasoning-related domains like coding, such extensions are not directly applicable in our setting. Methodologically, the injection of *trigger sentences* is ill-suited for code generation tasks, where inserting additional tokens may introduce syntactic or semantic errors. We therefore leave the integration of ID-sampling to future work.

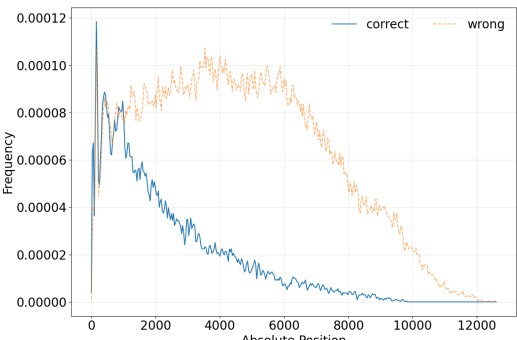

Figure 8: Positional frequency of thinking-related linguistic markers in Deepseek-R1-distill-Qwen-7B responses using ID-sampling on the AIME-2024 dataset.

## E    LINGUISTIC MARKERS

In the paper, we have presented the occurrence frequencies of the linguistic markers as our motivation. We provide the full list of such markers here, which is the same as Wang et al. (2025a): "however", "wait", "alternatively", "hmm".

## F    ADDITIONAL EXPERIMENT RESULTS

### F.1    GPQA RESULTS

| Model | Sampling | Cons@1 | Cons@4 | Cons@8 | Cons@16 | Cons@32 |
|---|---|---|---|---|---|---|
| Deepseek-R1-7B | Vanilla Sampling | 48.9 | 59.7 | 65.2 | 67.3 | 70.1 |
| | ID Sampling | **50.0** | **60.8** | **66.8** | **67.9** | **71.1** |
| Qwen3-8B | Vanilla Sampling | **55.4** | 65.2 | 71.1 | **73.9** | 74.4 |
| | ID Sampling | **55.4** | **66.8** | **72.2** | **73.9** | **76.6** |

Table 4: ID-sampling performance on difference models on GPQA-diamond dataset. Specifically, Deepseek-R1-7B denotes Deepseek-R1-distill-Qwen-7B.

In GPQA, the wall-clock time of ID-sampling and vanilla sampling differs by only about $1.10\times$, a negligible overhead compared to the inherent runtime randomness of LLMs. Accordingly, we do not report equivalent-$N$ results in this setting. As shown in Table 4, we observe that on both models, ID-sampling stably outperform vanilla sampling when combined with majority voting (Cons@n). We choose not to report Pass@N, as for multiple-choice questions such as those in GPQA, as this metric primarily incentivizes diversity in responses rather than better answer quality.

## G    FREQUENCY OF LINGUISTIC MARKER

In Fig. 8, we show the frequency of linguistic marker after using ID-sampling. Specifically, because of the smooth applied to the curve, there are only a few spikes that are noticable. Aside from these spikes, we do not observe significant changes in the overall shape of the frequency curves for either correct or incorrect trajectories when comparing vanilla sampling to ID sampling. Nonetheless, the average frequency for correct trajectories appears slightly higher, while the frequency for incorrect trajectories is roughly the same.

We believe the lack of major change in incorrect trajectories is due to the limited number of "wait" prompts added. And these added patterns convert more incorrect to correct trajectories, without significantly disrupting the overall generation patterns. Similarly, the slight increase in average frequency for correct trajectories is likely due to a subset of previously incorrect paths being corrected.

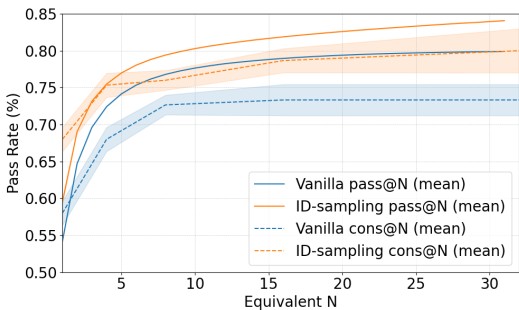

Figure 9: Performance of ID-sampling on Qwen3-8B evaluated on AIME-25. Results are averaged over five random seeds, with shaded regions indicating standard deviation.

This increases the overall average, as these corrected trajectories tend to exhibit more frequent occurrences of reasoning-related linguistic markers in their earlier segments.

## H  STATISTICAL SIGNIFICANCE

Due to the constraints of computational resources and the breadth of domains and models required to demonstrate the generalizability of our approach, we report results from a single run in the main paper. Here, we supplement those findings with an illustrative statistical evaluation using Qwen3-8B on AIME25.

As shown in Fig. 9, ID sampling consistently outperforms vanilla sampling. The standard deviation of Pass@n is extremely small and almost imperceptible in the plot because we use an unbiased estimator of Pass@n. Although the variance of Cons@n is larger, our proposed method remains consistently superior across runs.

