# OpenReview forum: "Efficient Test-time Scaling via Iterative Deepening"
_ICLR.cc/2026/Conference — Submitted to ICLR 2026_

### Official Review · Reviewer_1ofN · 2025-10-30

**Soundness:** 2
**Presentation:** 3
**Contribution:** 3
**Rating:** 6
**Confidence:** 4

**Summary:**

This paper proposes ID-Sampling, a test-time scaling method that inserts lightweight trigger sentences (e.g., *“Wait, let me reconsider.”*) at appropriate positions to induce self-correction during reasoning. The method is conceptually simple and model-agnostic. It introduces a trigger position predictor and performs multiple inference branches, selecting answers based on vote consensus. Experiments on AIME’24/25, GPQA, Math-500, and other reasoning benchmarks demonstrate that ID-Sampling improves pass@k accuracy under equivalent total sample budgets. The authors provide extensive analysis on linguistic markers, trigger placement, and interactions with model scale.

**Strengths:**

* **Originality**: While test-time scaling has been explored recently, the idea of using lightweight trigger insertions to induce self-correction is novel and orthogonal to simple sampling or self-consistency strategies. The paper highlights an interesting emergent behavior of modern LLMs.
* **Quality**: Empirical results are strong and consistent across diverse reasoning tasks. The *Equivalent N* setting is carefully designed to ensure fair comparison among sampling-based baselines.
* **Clarity**: The paper is clearly written, well-structured, and provides intuitive explanations of why the approach works. Figures illustrating trigger positions and linguistic markers further help understanding.

**Weaknesses:**

1. **Lack of variance and multi-run reporting.**
   Because reasoning evaluation with sampling is high-variance (especially on AIME-like tasks with only 30 questions), results should be averaged over multiple random seeds or include standard deviations / confidence intervals. The current reporting appears based on single runs, which weakens statistical reliability.

2. **Natural trigger generation is acknowledged but not analyzed.**
   The paper explicitly notes that the model may naturally produce trigger-like sentences *before* manual insertion. However, it does not quantify:

   * the frequency of such cases,
   * whether redundant triggering harms or saturates reasoning,
   * or how forced triggers interact with already-correct trajectories.
     This represents a missing failure-mode analysis.

3. **Limited exploration of the γ hyperparameter.**
   Although γ tunes allowed insertion positions, its scaling behavior is only partially analyzed. It is unclear whether larger γ values eventually hurt performance.

**Questions:**

1. **On experimental stability:**

   * Are results averaged over multiple inference runs / seeds?
   * If so, please report mean ± std or confidence intervals.
   * If not, can the authors comment on observed variance across seeds?

2. **On natural triggers:**
   Section 4 acknowledges that models sometimes spontaneously generate trigger markers (e.g., *“wait”*). Could the authors quantify:

   * how often this happens,
   * how it affects ID-Sampling performance,
   * and whether forced insertion becomes redundant or harmful?

3. **γ scaling behavior:**
   Have the authors explored performance when γ continues increasing beyond the reported values? Is there a tipping point where too many potential trigger positions reduce accuracy by distracting the model?


### **Additional Suggestions**

* The related work section is missing discussion on two highly relevant works on efficient test-time scaling:

  1. Inference scaling laws: An empirical analysis of compute-optimal inference for LLM problem-solving. ICLR 2025
  2. Every Rollout Counts: Optimal Resource Allocation for Efficient Test-Time Scaling. NeurIPS 2025

  Both works analyze compute allocation under sampling, which overlaps with the core motivation of this paper. Please discuss both works on Related work section.

---

> ### Author Response · Authors · 2025-11-20
>
> Thank you for your detailed review and helpful suggestions. We have updated the manuscript accordingly, and we invite you to review the revisions. Below, we highlight the changes and address your questions and concerns one by one.
>
> >  Lack of variance and multi-run reporting
>
> Thank you for the suggestion. **We have added an additional discussion of statistical significance in Section H in the appendix. ** Because our paper evaluates multiple models across multiple datasets, it is not feasible to conduct significance tests for every setting due to computational limits. However, we have included results for Qwen3-8B on AIME25 over five seeds. **The observed standard deviation is less than 0.1 for pass@n and around 2 for cons@n, indicating that our results remain statistically robust.**
>
> Besides, in our original paper, we already employed the unbiased estimator for pass@n [1], which accounts for randomness by sampling more responses and mathematically computing the expected pass rate.
>
>
> > Natural trigger generation is acknowledged but not analyzed.
>
> We would like to clarify that although we did not specifically analyze the natural trigger generation of the exact word “wait,” our work is motivated by the broader frequency patterns of linguistic markers related to reflection in correct responses. These markers include, but are not limited to, the word “wait.” The frequency of “wait” follows the same trend as other markers in this category, and therefore, we did not find it necessary to include additional figures that would be nearly identical to the current Figure 1.
>
> We have updated our paper draft to state this more clearly, highlighted in blue.
>
> > Have the authors explored performance when γ continues increasing beyond the reported values? Is there a tipping point where too many potential trigger positions reduce accuracy by distracting the model?
>
> In our current ablation study, we have already explored $\gamma$ in the range {1.5, 2.0, 2.5}. Our recommended setting is $\gamma = 2.0$. **Decreasing $\gamma$ (which increases the insertion frequency) or increasing $\gamma$ (which decreases the insertion frequency) both lead to degraded performance.** As discussed in the paper, this occurs because deviating in either direction fails to match the characteristic patterns of correct generations, which reduces the effectiveness of the proposed method.
>
> > The related work section is missing discussion on two highly relevant works on efficient test-time scaling
>
> Thank you for pointing out these relevant papers. We have added both to our related work section. The first paper you mentioned serves as an early exploration of test time scaling, playing a role similar to the originally cited Snell et al. The second paper proposes DORA, a test-time scaling algorithm that computes a weight based on the PRM score. Our algorithm does not require the use of a PRM, and therefore this work does not directly serve as a baseline for our method. However, both are relevant to our prior work section, and we appreciate pointers.
>
> [1]. Mark Chen, Jerry Tworek, Heewoo Jun, Qiming Yuan, Henrique Ponde De Oliveira Pinto, Jared Kaplan, Harri Edwards, Yuri Burda, Nicholas Joseph, Greg Brockman, et al. Evaluating large language models trained on code. arXiv preprint arXiv:2107.03374, 2021

---

### Official Review · Reviewer_haoU · 2025-11-01

**Soundness:** 3
**Presentation:** 2
**Contribution:** 2
**Rating:** 4
**Confidence:** 3

**Summary:**

The paper proposes a sampling technique that can further scale test-time computation for reasoning models and deliver better performance on reasoning-intensive benchmarks. They showed the theoretical result + empirical validation.

**Strengths:**

* empirical validation + ablation study
* short theory result

**Weaknesses:**

* presentation is a bit confusing, didn't show experiments until the end
* Captioning of FIgure 4/5/6 very difficult to parse
* improvement over baseline is quite small at k=16, which seems to suggest the improvement ultimately diminishes as k increases

**Questions:**

Do authors believe benefit of ID would scale as people choose larger k?

---

> ### Author Response · Authors · 2025-11-20
>
> Thank you for your constructive review. Here we answer to your concerns one by one:
>
> > presentation is a bit confusing, didn't show experiments until the end
>
> We are unsure what specific issue you are referring to. In our paper, we present all experimental results in a dedicated Experiments section, consistent with standard structure in prior work [1, 2]. Nevertheless, we have updated our paper to include one representative result in the early part of the manuscript. If you could clarify in follow up comments, we are happy to address and improve where needed clarity of the paper.
>
> > Captioning of FIgure 4/5/6 very difficult to parse
>
> Thank you for the suggestion. We agree that the original layout of Figures 4, 5, and 6 was tight, which made the captions harder to read. We have revised the manuscript to improve their clarity and spacing. We invite you to review the updated version, and we would be happy to make further refinements if needed.
>
> > Do authors believe benefit of ID would scale as people choose larger k?
>
> We would like to clarify that, relative to the number of samples kkk, the effectiveness of ID sampling depends more on whether the model has already saturated on the dataset. As shown in Figures 4, 5, and 6, when the models have bigger potential in test-time scaling, ID sampling continues to provide stable improvements even at k=32 since earlier models do not saturate when we reach k=32. We have additionally included results for k=64 on Qwen3-8B for AIME25 below, which still demonstrate a clear advantage for ID sampling.
> |                  | Pass@64 | Cons@64 |
> |------------------|---------|---------|
> | Vanilla-sampling | 80.06   | 80.00   |
> | ID-sampling      | 85.63   | 83.33   |
>
> **Overall, our findings show that ID sampling improves the test time scaling capabilities of LLMs, particularly large reasoning models, without requiring additional training or complex manipulation of the latent space. Even in scenarios where model performance saturates at smaller k, ID sampling can still reduce the number of samples needed to reach the saturation level.** For example, on AIME25, ID sampling yields up to a 2.5x reduction in the number of samples required for Qwen3 32B to reach the same pass@n, as shown in Table 2, and up to a 1.6x reduction for Qwen3 8B, as shown in Figure 5.
>
> [1]. Chen X, Aksitov R, Alon U, et al. Universal self-consistency for large language model generation[J]. arXiv preprint arXiv:2311.17311, 2023.
>
> [2]. Lee K H, Fischer I, Wu Y H, et al. Evolving deeper llm thinking[J]. arXiv preprint arXiv:2501.09891, 2025.

---

### Official Review · Reviewer_zWVa · 2025-11-02

**Soundness:** 3
**Presentation:** 3
**Contribution:** 2
**Rating:** 2
**Confidence:** 3

**Summary:**

The paper studies LLM behaviors with "longer reasoning" by inserting phrases like "wait" during reasoning to trigger rethinking behavior. Using three methods (BoN, Pass@N, Majority Voting) across three datasets MATH, AIME, GPQA the paper proves to be more effective than S1.

The method is effective on both reasoning/non-reasoning models tested on a diverse family of models Phi, Llama, R1-Distill, and Qwen3.

**Strengths:**

(1) The proposed method is supported by enough experiments that I am persuaded that this method of inserting "wait" will work in the way paper describes it.

**Weaknesses:**

(1) In terms of novelty I am not sure how helpful this is to the field. The fact that we can boost performance by simply inserting a few words have already been shown by S1. Though the paper differs to S1 in that it comes up with their own schedule based on their observations how llm reasoning emprically behaves, I dont think its novel enough.

(2) Some tables are misleading. For instance Table 1.b at N=32 both Vanilla and Id-Sampling scores 20.69 for cons@N, however the paper highlights Id-sampling only.

(3) We can easily see most of the performance gains are relatively small. Especially for Table2(AIME2025) the performance gap is mostly <6 points. Considering AIME only has 30 questions Im not sure if this is a statistically confident performance gain.

(4) While the paper has a short mention on the justification of their selection of the trigger sentence, it sounds extremely empirical. I wouldn't be surprised if their is a random word that just works better than 'wait'.

**Questions:**

See weaknesses.

---

> ### Author Response · Authors · 2025-11-20
>
> Thank you for your time and effort in reviewing our paper. We sincerely appreciate the thoughtful feedback. Below, we address each of your concerns in turn.
>
> > Though the paper differs to S1 in that it comes up with their own schedule based on their observations how llm reasoning emprically behaves, I dont think its novel enough.
>
> Thank you for acknowledging the novelty of our scheduling strategy for inserting “wait” tokens during the reasoning process. We would like to further emphasize that, unlike S1, whose primary objective is budget forcing, our method does not substantially alter the generation length. Thus, our method is more about changing the distribution of “wait” rather than purely inserting them.
> **Given the rapid development of LLMs, we believe that our algorithm, as a simple yet effective method, provides a high practical value to the community due to its ease of adoption.** We believe this also explains the appeal of prior works such as S1 [1] and DAPO [2], which similarly fall into the category of simple yet effective approaches, and hence, we believe our contribution will similarly be of service as a published paper for the community.
>
> > Some tables are misleading.
> Thank you for pointing this out. We acknowledge that we previously missed the bold in Table 1, and we have updated the manuscript accordingly. This correction does not affect our conclusion that ID-sampling is generally superior to vanilla sampling and S1. We are happy to further revise the paper if you identify any additional issues.
>
> > Considering AIME only has 30 questions, I'm not sure if this is a statistically confident performance gain.
>
> Thank you for pointing this out. We have added an additional discussion of statistical significance in Section H of the appendix. Because our paper evaluates multiple models across multiple datasets, conducting significance tests for every setting is computationally impractical. However, we have now included results for Qwen3 8B on AIME25 over five seeds. The observed standard deviation is less than 0.1 for pass@n and around 2 for cons@n, while the differences in mean pass@n are on the order of at least 1, and the differences in mean cons@n are at least 3 in most settings. **These margins indicate that our results remain statistically robust.**
>
> We would also like to clarify that our original submission already employed the unbiased estimator for pass@n [1], which accounts for randomness by sampling more responses and mathematically computing the expected pass rate.
>
>
> > I wouldn't be surprised if their is a random word that just works better than 'wait'.
>
> On one hand, the choice of the word “wait” is consistent with common practice in the field, with S1 being one representative example. Because LLM behavior evolves over time and the set of effective linguistic markers may also shift, **our work does not aim to identify the optimal marker. Instead, our focus is on whether adjusting the injection frequency of such markers can yield performance gains.**
>
> On the other hand, we agree with your point that alternative designs may further improve upon our current ID-sampling approach. However, we believe this does not undermine our contribution. Rather, it highlights that our algorithm represents a solid step toward understanding and improving test-time scaling. **The potential for additional enhancements should be viewed as a strength, not a limitation, since our method provides foundational building blocks for future work that improve upon vanilla sampling and S1.** Our results open promising directions for exploring whether mimicking other generation patterns from the correct responses can further boost performance.
>
> [1]. Muennighoff N, Yang Z, Shi W, et al. s1: Simple test-time scaling[C]//Proceedings of the 2025 Conference on Empirical Methods in Natural Language Processing. 2025: 20286-20332.
>
> [2]. Yu Q, Zhang Z, Zhu R, et al. Dapo: An open-source llm reinforcement learning system at scale[C]. Neurips, 2025.

---

> > ### Comment · Reviewer_zWVa · 2025-11-23
> >
> > Thanks for the update. Your clarifications on (4) helped me better understand the paper. I also appreciate the update on (2) and (3).
> >
> > Two more questions
> >
> > (1) Recent papers like "Does Reinforcement Learning Really Incentivize Reasoning Capacity in LLMs Beyond the Base Model?" argue that RL does not unlock new capabilities; instead, it's more about improving sample efficiency. Based on such, where does the author think the margin comes from? Is the ID-sampling enhancing sample efficiency? or is it unlocking something new?
> >
> > (2) In terms of generation cost, does ID-sampling require more tokens than plain decoding?

---

> > > ### Author Response · Authors · 2025-11-23
> > >
> > > We thank the reviewer for the continued discussion and are glad that our previous responses have addressed your concerns. Below, we respond to your additional questions:
> > >
> > > >  where does the author think the margin comes from?
> > >
> > > We are unsure what specific concern you are referring to here. As indicated by our title, our paper focuses on test time scaling, which is distinct from RL. We believe that the potential connection arises because a test time scaling algorithm, particularly a sampling algorithm such as ours, can serve as the rollout policy within an RL framework. In such a setting, ID-sampling can lead to a higher pass@n at test time, which raises the upper bound of what an RL policy trained with ID-sampling can potentially achieve compared to one trained with plain decoding.
> > >
> > > Separately from this upper bound consideration, to reach the same upper bound, ID-sampling requires fewer samples and therefore fewer rollouts in RL, meaning fewer responses need to be generated per prompt in each batch. This in turn improves training efficiency.
> > >
> > > The reason for this improvement is that ID-sampling shifts the generation distribution toward trajectories that exhibit correct reasoning patterns. Although plain sampling can in principle produce similar responses, the probability of sampling them is significantly lower.
> > >
> > > > does ID-sampling require more tokens than plain decoding?
> > >
> > > This depends on the model used in the setting. This is precisely what motivates our introduction of the idea of equivalent N in our paper, which incorporates the additional total computation time into the comparison. As noted in lines 359-364 and lines 413-420, for reasoning models, which are our primary focus, ID-sampling requires up to 1.3 times the computation cost compared to plain decoding. However, as shown in our experiments, ID-sampling still outperforms plain decoding even when this overhead is taken into account.
> > >
> > > Thank you again for your time and attention to our paper. We are happy to answer any further questions you may have, and we would greatly appreciate it if you would consider raising your score if you have no remaining concerns.

---

> > > > ### Comment · Reviewer_zWVa · 2025-11-24
> > > >
> > > > Ive updated my score to 4.

---

> > > > > ### Author Response · Authors · 2025-11-25
> > > > >
> > > > > Dear reviewer,
> > > > >
> > > > > Thank you for updating the score. We would like to follow up to better understand any remaining concerns that may be preventing you from updating your score to one above the borderline.
> > > > >
> > > > > Regarding novelty, which was the only point in your initial list of weaknesses that was not explicitly mentioned in your first response, we would like to reiterate that, unlike S1, **our paper is the first to show that test-time scaling can be improved by mimicking the generation patterns found in correct responses.** We believe this finding provides substantial value to the ICLR community, as it highlights a promising direction for future research: identifying systematic differences between correct and incorrect responses and leveraging these differences in generation patterns to improve model performance. **Our work offers a practical demonstration of how such differences can be used effectively, opening a broad space for future exploration.**

---

### Author Response · Authors · 2025-11-20
**General Response**

We thank all reviewers for their insightful feedback. In addition to addressing the specific questions raised, we have updated the manuscript to include statistical tests of our results in the appendix and have made minor layout adjustments as recommended by Reviewer haoU. All changes are highlighted in blue where possible. We invite you to review the updated version, and we are happy to respond to any further questions or concerns.

Thank you again for your time. We would greatly appreciate it if you could let us know whether our rebuttal has addressed your concerns and consider updating the scores accordingly.

---

### Meta-Review · Area_Chair_grkF · 2026-01-05

**Summary:**

This paper demonstrates that including words like "wait" that increase the model's tendency to reflect and mimic the patterns of successful generations tends to increase performance. This leads to a simple test-time intervention to improve model performance.

**Reviewer Concerns:**

Author's concerns surrounded novelty, and the specificity with which the authors were able to pinpoint the mechanism  behind their findings. While the work still makes a novel contribution, the reviewers still felt relatively unenthusiastic about the submission. Other specific concerns were addressed by the authors.

**Reviewer Scores:**

One reviewer raised their score to a 4, and the other I doubt would have increased beyond a six; this lead to a review somewhat below the bar for acceptance. My sense is that the reviewers find the contribution unsurprising, and their enthusiasm does not merit this crossing the par for acceptance given the selective nature of the conference.

---

### Decision · Program_Chairs · 2026-01-26

Reject